# Critical Protein–Protein Interactions Determine the Biological Activity of Elk-1, a Master Regulator of Stimulus-Induced Gene Transcription

**DOI:** 10.3390/molecules26206125

**Published:** 2021-10-11

**Authors:** Gerald Thiel, Tobias M. Backes, Lisbeth A. Guethlein, Oliver G. Rössler

**Affiliations:** 1Department of Medical Biochemistry and Molecular Biology, Saarland University Medical Faculty, D-66421 Homburg, Germany; maxi.backes9@googlemail.com (T.M.B.); oliver.roessler@uks.eu (O.G.R.); 2Department of Structural Biology and Department of Microbiology & Immunology, School of Medicine, Stanford University, Stanford, CA 94305, USA; libbyg@stanford.edu

**Keywords:** c-Fos, Egr-1, histone deacetylase, MAP kinase, mediator, SRF, SUMO

## Abstract

Elk-1 is a transcription factor that binds together with a dimer of the serum response factor (SRF) to the serum-response element (SRE), a genetic element that connects cellular stimulation with gene transcription. Elk-1 plays an important role in the regulation of cellular proliferation and apoptosis, thymocyte development, glucose homeostasis and brain function. The biological function of Elk-1 relies essentially on the interaction with other proteins. Elk-1 binds to SRF and generates a functional ternary complex that is required to activate SRE-mediated gene transcription. Elk-1 is kept in an inactive state under basal conditions via binding of a SUMO-histone deacetylase complex. Phosphorylation by extracellular signal-regulated protein kinase, c-Jun N-terminal protein kinase or p38 upregulates the transcriptional activity of Elk-1, mediated by binding to the mediator of RNA polymerase II transcription (Mediator) and the transcriptional coactivator p300. Strong and extended phosphorylation of Elk-1 attenuates Mediator and p300 recruitment and allows the binding of the mSin3A-histone deacetylase corepressor complex. The subsequent dephosphorylation of Elk-1, catalyzed by the protein phosphatase calcineurin, facilitates the re-SUMOylation of Elk-1, transforming Elk-1 back to a transcriptionally inactive state. Thus, numerous protein–protein interactions control the activation cycle of Elk-1 and are essential for its biological function.

## 1. Introduction

The human genome encodes more than 1500 transcription factors. These proteins have a modular structure with distinct domains for DNA binding and transcriptional activation (or repression). Stimulus-responsive transcription factors connect cellular stimulation with gene transcription, involving protein kinases as signal transducers. These transcription factors contain phosphorylation-responsive activation domains, that is, transcriptional activation requires phosphorylation. In this article we discuss the molecular mechanism of the transcriptional activation of the stimulus-responsive transcription factor Elk-1, which is essentially involved in the control of cellular proliferation induced by mitogens, in thymocyte development, glucose homeostasis and brain function. Elk-1 is a member of the ETS family of transcription factors. The name ETS is derived from the avian erythroblastosis virus E26, which encodes the v-*ets* oncogene. Its cellular counterpart, Ets-1, is the founding protein of the ETS family of proteins. All family members bind to a similar DNA recognition site, encompassing the sequence GGAA/T, using their DNA binding site, the Ets domain [1]. In this review, we focus on the essential role of numerous protein–protein interactions for the activation of Elk-1, involving several protein kinases, a Ca^2+^/calmodulin-dependent protein phosphatase, chromatin modifiers and the transcription factor SRF, which all work in a sequential manner.

## 2. The Serum-Response Element

Cells express receptors and ion channels to enable communication with their environment. The stimulation of cells with hormones, neurotransmitters, cytokines, mitogens, nutrients or metabolites can change their genetic program. This stimulus-regulated gene transcription allows cells to respond to extracellular stimuli that are released by other cells of the organism. Protein kinases function as signal transducers, leading to the activation of stimulus-responsive transcription factors that bind to specific genetic elements. One of these elements is the serum-response element (SRE) (Figure 1A), which facilitates gene transcription as a result of cellular stimulation by extracellular signaling molecules. The SRE is the binding site for the serum response factor (SRF) and the transcription factor Elk-1 or related ternary complex factors (TCFs) (Figure 1A) [1,2]. An SRF dimer binds to the CArG box with the consensus sequence CC[A/T]_6_GG, while the TCF proteins interact with the Ets consensus core sequence GGAA/T, typical for the transcription factors of the ETS family. It has been suggested that the binding of Elk-1 to DNA involves phosphorylation [3]. However, recent chromatin immunoprecipitation (ChIP) data showed that unphosphorylated Elk-1 binds to DNA under basal conditions [4,5]. Likewise, the generation of the ternary complex between Elk-1 and SRF occurs independent of Elk-1 phosphorylation [6].

## 3. Modular Structure of Elk-1 and Other Ternary Complex Factors

The modular structures of Elk-1 and related ternary complex factors are depicted in Figure 1B. The proteins share a similar domain structure, encompassing an N-terminal DNA binding domain (A-box) and a C-terminal transcriptional activation domain (C-box). The B domain which lies between the DNA binding and activation domains functions as a “grappling hook” (M. Treisman) for the formation of the ternary Elk-1-SRF complex. Thus, Elk-1 contacts a particular DNA sequence and binds to a dimer of the SRF in order to generate a ternary complex. The three TCF proteins are ubiquitously expressed in many cell types [7,8]. However, in embryonic stem cells Elk-1 expression is significantly higher than the expression of SAP-1 and SAP-2 [9]. Elk-1 is a major nuclear substrate for stimulus-responsive MAP kinases that phosphorylate serine residues S383 and S389 within the transcriptional activation domain [10,11,12,13,14,15,16]. The phosphorylation of Elk-1 is induced by the stimulation of the cells with ligands for G protein-coupled receptors, receptor tyrosine kinases, cytokines or by the activation of ligand-gated and voltage-gated Ca^2+^ channels that activate stimulus-responsive protein kinases in the cell. Elk-1 connects these intracellular signaling cascades with SRE-dependent gene transcription (Figure 2). Phosphorylated Elk-1 is dephosphorylated by the Ca^2+^/calmodulin-dependent protein phosphatase calcineurin, which functions as a negative feedback regulator of the stimulus-induced phosphorylation and activation of Elk-1.

## 4. Tools to Investigate Elk-1 Activity

The analysis of transgenic mice with a disruption of one of the TCF-encoding genes revealed that the TCF proteins Elk-1, SAP-1 and SAP-2 share redundant functions resulting from the structural homology between the TCF proteins [17,18,19,20]. Therefore, gene targeting technology used to inactivate a single TCF-encoding gene was not useful for the investigation of TCF function. Additionally, double-transgenic Elk-1^−/−^/SAP-1^−/−^ mice are infertile [20]. It is therefore difficult to generate adult triple-transgenic Elk-1^−/−^/SAP-1^−/−^/SAP-2^−/−^ mice for assessing the role of TCF proteins in different tissues using standard gene-targeting techniques and mice breeding.

The problem associated with the redundancy of functions between TCF proteins was solved by the design and expression of a dominant-negative mutant of Elk-1, REST/Elk-1ΔC (Figure 3A). The mutant retains the domains for binding to DNA and to SRF, but lacks the phosphorylation-regulated activation domain. REST/Elk-1ΔC blocks the binding sites of TCF proteins to DNA and interferes with the interaction of TCF proteins with SRF because of the presence of the B domain in the Elk-1 mutant (Figure 3B). Additionally, the expression of the truncated Elk-1 protein with a transcriptional repression domain, derived from the transcriptional repressor REST, allows the recruitment of histone deacetylases to the transcription units bound by REST/Elk-1ΔC, inducing the chromatin compaction of these sites.

## 5. Biological Functions of Elk-1

Elk-1 regulates the proliferation and apoptosis of several cell types, including astrocytes, fibroblasts, and pancreatic β-cells [8,21,22]. In primary astrocytes, the expression of the dominant-negative Elk-1 mutant REST/Elk-1ΔC completely blocks epidermal growth factor (EGF)-induced proliferation of the cells [21] (Figure 4A), indicating that Elk-1 activity is required for the mitogenic activity of EGF. The expression of REST/Elk-1ΔC in pancreatic β-cells of transgenic mice revealed that transgene expression resulted in the generation of significantly smaller islets (Figure 4B) [8]. Thus, an expression program involving Elk-1 or related TCF proteins is essential for the generation of islets of normal size. Expression of the dominant-negative mutant of Elk-1 strongly activates caspase-3/7 (Figure 4C), enzymes that play critical roles in the execution phase of apoptosis. Thus, the activation of apoptosis is responsible for the generation of smaller islets in transgenic mice expressing the dominant-negative mutant of Elk-1 in pancreatic β-cells. As a result, the transgenic mice showed impaired glucose tolerance [8] (Figure 4D). Elk-1 activity has additionally been connected with the regulation of adipogenesis, thymocyte development and brain function [17,20,23,24]. Several proteins of the ETS family of transcription factors have been linked to tumor development and cancer [1]. The expression of v-*ets*, encoded by the E26 virus, results in the development of erythroid and myeloid leukemias. The ETS proteins PEA3 and ERG have been proposed to induce Ewing sarcomas and prostate cancer, respectively. Elk-1 has been proposed to play a role in different cancer types. However, the published studies do not allow for a general statement to be made regarding a correlation between Elk-1 and tumor development.

## 6. Essential Protein–Protein Interactions of Elk-1

### 6.1. Generation of a Ternary Complex with SRF: Role of DNA Binding and Protein–Protein Interactions

Elk-1 and SRF are transcription factors that bind in a sequence-specific manner to cognate DNA sites. Point mutations of the Elk-1 DNA binding site and the SRF DNA binding site were used to determine the importance of DNA binding in general and of each binding site in particular. The SRE derived from the c-Fos gene, and mutations leading to an inactivation of Elk-1 DNA binding (ΔEts) or SRF DNA binding (ΔCArG) are shown in Figure 5A. Stimulation of the ERK1/2 signaling pathway via activation of a B-Raf mutant induces c-Fos-promoter-regulated gene transcription. Mutational inactivation of either the Elk-1 or the SRE DNA binding site leads to a significant reduction of SRE-mediated transcription [25]. Thus, the DNA binding of both Elk-1 and SRF is required for B-Raf-induced gene transcription via the SRE. SRE-mediated transcription is also activated by stimulating transient receptor potential melastatin-3 (TRPM3) channels with the ligand pregnenolone sulfate, mediated by an influx of Ca^2+^ into the cells (Figure 2). TRPM3 channels function as a noxious heat sensor in somatosensory neurons and have been described to regulate insulin secretion [26,27]. TRPM3 stimulation activates the c-Fos promoter. Mutation of the DNA binding site for SRF significantly impairs TRPM3-induced activation of the c-Fos promoter, while mutation of the TCF DNA binding site (ΔEts) has no effect on TRPM3-induced upregulation of gene transcription. Experiments with the dominant-negative mutant of Elk-1 have shown that TRPM3-induced reporter gene transcription is reduced in the presence of the Elk-1 mutant [28]. These data indicate that Elk-1 is involved in TRPM3-induced gene transcription, although DNA binding is not essential. We believe that Elk-1 is recruited to the SRE via protein–protein interaction with SRF, which would not require DNA binding by Elk-1. Likewise, the activation of SRE-regulated gene transcription following the stimulation of G protein-coupled receptors was shown to require the DNA binding of SRF, but was independent of the DNA binding site of Elk-1 [29,30,31]. Thus, SRE-regulated transcription requires the DNA binding of SRF, which then recruits Elk-1 and other TCFs to the SRE. Elk-1 binds to DNA and SRF or only to SRF via protein–protein interaction through its B-domain.

The B-domain of Elk-1 and other TCF proteins is necessary and sufficient for the interaction with SRF. Deletion of the entire domain or mutation of a critical threonine residue (T163) within the B-domain abolishes SRF binding [32]. This threonine residue is also conserved between the three homologous TCF proteins (Figure 5B). Alanine scanning mutations of the Elk-1 B-box identified four hydrophobic residues (Y153, Y159, F162 and I164) involved in SRF binding [33]. These sites are also conserved in SAP-1 and SAP-2, except for isoleucine residue 164 which is replaced by a leucine residue in SAP-1. An X-ray structural analysis of the SAP-1 B-Box together with SRF showed that the hydrophobic residues Y141, L246, F150 and L155 of SAP-1 contribute to the binding interface to SRF [34]. All four sites are conserved in Elk-1 and SAP-2, and the corresponding SAP-1 residues Y141 and F150 (Y153 and F162 in Elk-1) have also been identified via alanine scanning mutation as necessary for SRF binding [33]. The X-ray structure of the SAP-1 B-box revealed that this domain adopts a 3_10_-helix/β-strand/3_10_-helix conformation that allows the interaction to SRF with a hydrophobic interface [34].

### 6.2. Phosphoacceptor Motifs of Elk-1

ChIP experiments showed that unphosphorylated Elk-1 occupies the SRE of the c-Fos and the Egr-1 promoters in the absence of stimulation. Following mitogenic stimulation, DNA-bound phosphorylated Elk-1 is detectable 10 to 15 min after stimulation [4,5]. The TCF proteins are substrates for three MAP kinases extracellular signal-regulated protein kinases (ERK1/2), c-Jun N-terminal protein kinase (JNK) and p38 protein kinase. These kinases form a family of structurally related protein kinases that are integrated into protein kinase cascades that include MAP kinase kinases (MAP2 kinases) and MAP kinase kinase kinases (MAP3 kinases). MAP kinases are activated by numerous signaling molecules, including ligands of G protein-coupled receptors, receptor tyrosine kinases, cytokine receptors and various stress-induced agents such as osmotic or oxidative stress and UV radiation [35,36,37].

MAP kinases are proline-directed serine/threonine kinases and several S-P or T-P clusters are present within the transcriptional activation domains of Elk-1, SAP-1 and SAP-2 which show high sequence conservation (Figure 6). Elk-1, SAP-1 and SAP-2 are all phosphorylated by ERK1/2 [7,38]. The serine residues S383 and S389 of Elk-1 have been identified as major phosphoacceptor sites for ERK2 and JNK [10,11,12,13,14,15,16]. In particular, the phosphorylation of S383 is important because mutation of this residue attenuates the transactivation function of Elk-1 more than mutations of other serine or threonine residues [11,12,14,16]. Kinetic experiments revealed that S383 and T368 of Elk-1 are phosphorylated very rapidly by ERK2, whereas residues T353, T363 and S389 show intermediate phosphorylation kinetics. ERK2 additionally phosphorylates serine and threonine residues outside of the C-Box (T336, T417, S423) [39]. JNK and p38 protein kinases phosphorylate Elk-1, with differences for site preferences and kinetics being observed. In particular, JNK shows a kinetic preference for T-P motifs, while p38 exhibits preferences similar to ERK2 [39]. The residues S381 in SAP-1 and S357 in SAP-2 (corresponding to the phosphorylation site S383 in Elk-1) have been shown to be important for the transcriptional activation of SAP-1 and SAP-2 [7]. Likewise, UV-radiation-activated JNK phosphorylates Elk-1 on serine residues 383 and 389, and JNK-mediated transcriptional activation was abolished when these residues were mutated to alanine [14]. ERK, JNK and p38 phosphorylate SAP-1 and SAP-2, although different results have been published about the phosphorylation of SAP-1 by JNK [15,40,41,42]. The p38 protein kinase phosphorylates both Elk-1 and SAP-1 on residues within the C-terminal activation domain, and functions as signal transducer connecting cytokine stimulation and TCF activation [15,43].

### 6.3. Docking Sites for Stimulus-Responsive MAP Kinases

Protein kinases have been characterized by their recognition of a particular consensus site which includes the phosphorylated residue. Often, short sequence motifs are recognized (e.g., the S/T-P site for the proline-directed MAP kinases). These motifs are often found several times in a protein, indicating that the recognition sequence alone cannot provide protein kinase specificity. Moreover, MAP kinases recognize similar consensus sites. Thus, the interaction of MAP kinases with other sites of the substrate proteins must occur in order to increase substrate affinity and provide protein kinase specificity. Elk-1 has at least two protein kinase docking sites, the D-sites (or D-domain, D-Box) and the DEF site. The D-domain encompasses a stretch of basic residues, a central proline residue and a cluster of hydrophobic residues known as the docking groove [35] (Figure 7). These sites are either conserved in the SAP-1 and SAP-2 molecules or replaced by similar charged residues (exchange of a lysine residue for an arginine residue). The D-box also functions as an ERK2 docking site in a heterologous context [44,45]. Activation of ERK2 is required for efficient binding to the D-box of Elk-1 [44]. ERK2 and JNK bind to similar and distinct amino acids of the D-box of Elk-1. Alanine scanning mutations underlined the importance of the basic amino acids R314 and K315 of Elk-1 N-terminal of the proline residue P316 and the residues R317 and L319 C-terminal of this proline residue for the binding of ERK2 and JNK. Mutation of the LEL cluster (amino acids 319–321) significantly reduced the docking function of the D-box [46]. Mutation of residues L323 and S324 revealed an impairment of JNK-induced Elk-1 phosphorylation. Additionally, exchange of the leucine residues 327 and 328 of Elk-1 with alanine residues completely blocked ERK2- and JNK-mediated Elk-1 phosphorylation, suggesting that these residues are also involved in MAP kinase docking. The leucine residues are not conserved in SAP-1 and SAP-2 but replaced by other hydrophobic amino acids (Figure 7). Figure 7 depicts for comparison the JNK binding site of the JNK-interacting protein JIP-1, showing that the N-terminal part of the D-box, including the stretch of basic amino acids, the proline residue and the cluster of hydrophobic amino acids, can be found in this protein as well [45]. In contrast to ERK2 and JNK, the p38 protein kinase does not use the D-box to bind to Elk-1 [45].

The second MAP kinase docking site is the DEF site, named after “docking site for ERK, FXF” [46]. Within the Elk-1 protein, the sequence FQFP is found C-terminal to the transcriptional activation domain. The motif FXFP is an evolutionarily conserved ERK docking site and functions independently of the D-box in Elk-1 protein. Likewise, ERK interacts with the D-box even when the DEF site is mutated [46]. SAP-1 and SAP-2 contain a DEF site as well (Figure 7). The DEF site functions as docking site for ERK1/2 and p38 protein kinases, while JNK does not interact with this site [39,45,46,47].

The fact that ERK interacts with Elk-1 involving two docking sites, the D-box and the DEF site indicates that ERK binds with higher affinity to Elk-1 than JNK or p38, which use either the D-box (JNK) or the DEF site (p38) for interaction. These different docking modules function independently and additively and result in different binding affinities for MAP kinases with a particular substrate [46]. Moreover, the docking site influences the phosphorylation site in proteins with multiple potential phosphorylation consensus sites. The binding of ERK to the DEF site directs phosphorylation of S383 in Elk-1, while binding of the kinase to the D-box results in the phosphorylation of other serine and threonine residues of Elk-1 [48].

### 6.4. Phospho-Elk-1 Interacts with the Mediator Complex

Experiments with cells derived from knockout mice revealed that expression of the mediator of RNA polymerase II transcription (Mediator) subunit 23 (Med23) is essential for stimulus-induced biosynthesis of the transcription factor Egr-1 [49,50]. The Mediator multiprotein complex consists of 20–30 subunits that function as a bridge between DNA-bound transcription factors, RNA polymerase II and general transcription factors. It transduces the signal from DNA-bound transcriptional activators to the transcriptional machinery [51]. As Egr-1 expression is controlled by Elk-1 or related transcription factors, the control of Elk-1 transcriptional activity by Med23 was investigated in the Berk lab. The transcriptional activation potential of Elk-1 was almost completely blocked in stimulated ES cells lacking Med23, while the expression of Med23 restored Elk-1 activity. Med23 bound specifically to phosphorylated Elk-1 [49], thus connecting Elk-1 phosphorylation with gene transcription. These data were corroborated in chromatin immunoprecipitation experiments showing that serum-induced phosphorylation of Elk-1 upregulates Mediator binding to the Egr-1 promoter region in ES cells, while this interaction is almost abolished in ES cells lacking Med23 [50]. A hydrophobic region within the transcriptional activation domain of Elk-1, including the residues F378 and W379, functions as docking site for Med23. Mutation of these residues to alanine residues completely abolishes the stimulus-induced transcriptional activation of Elk-1 [7]. However, the interaction of Med23 with the hydrophobic region of Elk-1 is not sufficient for binding, as shown in coprecipitation experiments of Elk-1 and Med23 [49]. Med23 additionally interacts with phosphorylated residues within the activation domain of Elk-1, leading to a stabilization of the protein–protein interaction and to an increased recruitment of RNA polymerase II to the transcription unit [9]. A recent analysis of multi-site phosphorylation of Elk-1 revealed that Elk-1–Mediator interaction was promoted by the phosphorylation of the fast and intermediate phosphoacceptor sites, while phosphorylation of the slow phosphorylation sites located outside of the activation domain counteracted Elk-1 activity by inhibiting Mediator recruitment [39]. Thus, the sequential phosphorylation of Elk-1 induces an activation of Elk-1, and later induces an inhibition of Elk-1, controlled by the interaction of phospho-Elk-1 with the Mediator complex. In addition, fully phosphorylated Elk-1 may additionally recruit the transcriptional repressor mSin3A to block gene transcription. The analysis of TCF–Mediator interaction further revealed that the TCF proteins interact differently with Med23. While Med23 is essential for Elk-1-regulated gene transcription, the transcriptional activation potential of SAP-2 is only marginally affected by the loss of Med23 [9], suggesting that other Mediator subunits connect SAP-2 with RNA polymerase II and the general transcriptional apparatus.

### 6.5. Protein–Protein Interaction of Elk-1 with the Histone Acetyltransferase p300

The transcriptional coactivators p300 and CREB binding protein (CBP) are involved in the regulation of many transcription factors, including CREB, nuclear hormone receptors, p53, c-Jun, c-Fos and many others [52]. CBP and p300 are histone acetyltransferases that catalyze the transfer of acetyl groups derived from acetyl-CoA to lysine residues of histones or other proteins. The acetylation of histones H3 and H4 is correlated with an open chromatin structure, facilitating the binding of transcription factors and RNA polymerase II to DNA. Elk-1 interacts with p300, even in the absence of stimulation (and phosphorylation). However, the phosphorylation of Elk-1 on serine residues S383 and S389 increases the affinity of Elk-1 to p300, which involves an additional interaction site within the p300 molecule. Moreover, the intrinsic histone acetyltransferase activity of p300 is dramatically increased. Mutation of the serine residues S383 and S389 to alanine prevents this increase in histone acetyltransferase activity [53]. Thus, the phosphorylation of Elk-1 induces the transcriptional activation by the recruitment of the Mediator complex and by interaction with the p300 coactivator, which leads to changes in the chromatin structure. In mouse embryonic fibroblasts it has been shown that the generation of these chromatin modifications requires phosphorylated and transcriptionally active Elk-1 [6]. p300 functions as bridge between Elk-1 and the general transcriptional machinery.

### 6.6. Calcineurin Catalyzes Dephosphorylation and Inactivation of Elk-1

The stimulus-induced upregulation of a biological activity requires a shut-off mechanism. Naturally, since Elk-1 is activated by phosphorylation, dephosphorylation is connected with the inactivation of Elk-1-mediated gene transcription. Kinetic studies have shown that the EGF-induced phosphorylation of Elk-1 on serine residue S383 is already strongly visible 15 min after stimulation, whereas after 60 min much lower levels of phospho-Elk-1 are detected [5,53]. The dephosphorylation and inactivation of phosphorylated Elk-1 is catalyzed by the Ca^2+^/calmodulin-dependent protein phosphatase calcineurin [54,55]. Specifically, calcineurin catalyzes the dephosphorylation of S383 of Elk-1 [55]. Calcineurin consists of the catalytic subunit A and the regulatory subunit B. Calcineurin A has a long catalytic domain and binding sites for calcineurin B and calmodulin (Figure 8A). An autoinhibitory domain is located on the C-terminus, which binds to the catalytic center in the absence of Ca^2+^ ions. A rise in intracellular Ca^2+^ activates calcineurin [56]. Experimentally, calcineurin can be activated by the expression of a truncated calcineurin A mutant (ΔCnA) (Figure 8A). The expression of ΔCnA significantly reduces the transcriptional activation potential of Elk-1 in cells expressing an activated G protein-coupled designer receptor (Rαq). Likewise, ΔCnA expression reduces the biosynthesis of the Elk-1-regulated target gene c-Fos (Figure 8B) [57]. ΔCnA expression also reduces the promoter activities of Egr-1 and c-Fos [57,58]. In the calcineurin-regulated transcription factor NFAT, a short binding site for calcineurin has been found, encompassing the sequence PxIxIT [59]; however, no such site is present in the Elk-1 molecule. Thus, the mechanism by which calcineurin binds to phospho-Elk-1 is currently unknown.

### 6.7. Recruitment of the mSin3A–Histone Deacetylase Complex to Highly Phosphorylated Elk-1

A quantitative analysis of multisite phosphorylation of Elk-1 revealed that phosphorylation of the “slow” sites T336, T417 and S422 of Elk-1 counteracted transcriptional activation and interaction with the Mediator complex [39]. These data indicate that the duration of Elk-1 activation is limited by progressive ERK2-catalyzed phosphorylation. The molecular mechanism of this inhibition lies in the recruitment of the corepressor mSin3A, a multiprotein complex which binds to the N-terminal repression domain of Elk-1 (amino acids 1–93), which overlaps with the DNA binding domain (Figure 9). This recruitment of mSin3A is stimulated by the phosphorylation of Elk-1 and correlates with the downregulation of c-Fos expression, encoded by a prominent target gene of Elk-1 [60]. Thus, there is a temporal delay between the fast phosphorylation of Elk-1 and the recruitment of the mSin3A complex. mSin3A has been described to be associated with histone deacetylase activity. Accordingly, pharmacological experiments using trichostatin A (TSA), an inhibitor of histone deacetylases, suggest that mSin3A recruits histone deacetylases to Elk-1. These enzymes catalyze the removal of the acetyl groups from histones, allowing ionic interactions between positively charged amino acids of histones and the negatively charged phosphate groups of the DNA. This results in a compaction of the chromatin structure, which is accompanied by a repression of gene transcription. Interestingly, binding of the histone deacetylase HDAC-1 to Elk-1 was shown to be enhanced following activation of the ERK signaling pathway [54].

### 6.8. A SUMO–Histone Deacetylase Complex Binds to Elk-1 in the Absence of Stimulation

Elk-1 contains a second portable repressor domain, the R motif (Figure 9), that functions in reducing the transcriptional activity of Elk-1 under basal conditions [61]. The Elk-1-related proteins SAP-1 and SAP-2 lack this R motif, but have other distinct repression domains. Within the R motif of Elk-1 there are two lysine residues K230 and K249 that function as acceptor sites for the ubiquitin-like protein SUMO, which is conjugated to the lysine residues of proteins involving an activating enzyme (E1), a conjugating enzyme (E2) and a ligase (E3). This modification is responsible for the repression activity of the R motif and is overcome by mutating the lysine residues K230 and K249 to arginine residues [62]. The lysine residues are part of the SUMO consensus site ψKxE (ψ, hydrophobic amino acid; x, any amino acid). In addition, acidic residues C-terminal to the SUMO consensus site are involved in SUMO binding, probably by enhancing binding of the SUMO-conjugating enzyme UBC9 [63]. The repression activity mediated by the R motif is TSA sensitive, indicating that SUMO functions via processes catalyzed by histone deacetylases. In fact, recruitment of the histone deacetylase HDAC-2 to Elk-1 occupied promoters has been shown [62,63]. These data indicate that the transcriptional repression of Elk-1 is mediated by the SUMO-dependent recruitment of histone deacetylases, which induces a chromatin compaction of Elk-1-bound genes. The phosphorylation of Elk-1 results in a loss of SUMO and HDAC-2 binding and therefore induces a de-inhibition of Elk-1 activity [62,63,64]. After phosphorylation and activation, Elk-1 is dephosphorylated, which allows its re-SUMOylation. Together, the R motif functions in keeping Elk-1 transcriptionally inactive under basal conditions.

## 7. Conclusions

Elk-1 is a DNA binding protein that interacts with a genetic element termed the “serum response element”. Biochemical and genetic investigation showed that the biological activity of Elk-1 is controlled by numerous protein–protein interactions. In fact, protein–protein interaction with the serum response factor may replace Elk-1–DNA interaction in some signaling pathways. Elk-1 is kept in an inactive state under basal conditions via the binding of a SUMO–histone deacetylase complex. A model has been formulated according to which Elk-1 exists under basal conditions as a “transcriptionally poised complex” [59] that is rapidly activated by cellular stimulation. The activation of MAP kinases, particularly ERK1/2, triggers a de-SUMOylation of Elk-1, leading to the phosphorylation and activation of Elk-1. The transcriptional activation potential of phosphorylated Elk-1 is enhanced by the interaction with subunits of the Mediator complex and binding of the histone acetyltransferase p300, leading to changes of the chromatin architecture. Strong and extended phosphorylation of Elk-1 attenuates Mediator recruitment and transcriptional activation by attracting the mSin3A–histone deacetylase complex. Moreover, phosphorylated Elk-1 is dephosphorylated by the Ca^2+^/calmodulin-dependent protein phosphatase calcineurin. Dephosphorylated Elk-1 is then re-SUMOylated to put Elk-1 back into an inactive state, waiting for the next cycle of stimulation and activation (Figure 10).

## Figures and Tables

**Figure 1 molecules-26-06125-f001:**
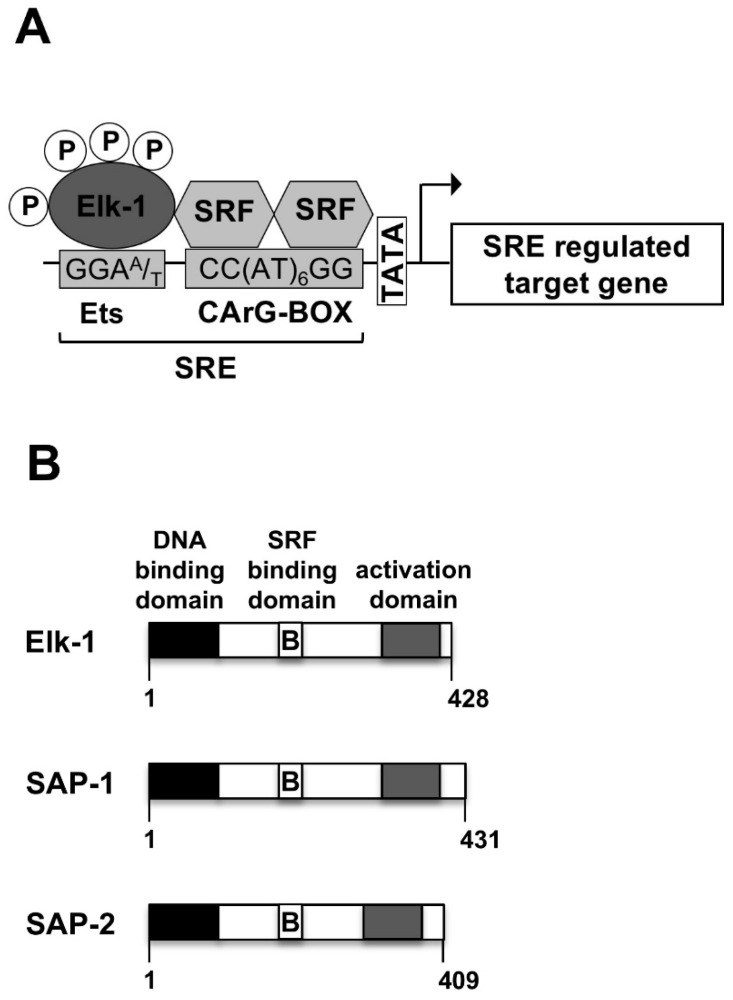
The serum-response element (SRE) provides binding sites for the serum response factor (SRF) and Elk-1. (**A**) The SRE. (**B**) Modular structure of Elk-1 and related ternary complex factors, showing the N-terminal DNA binding domain, the C-terminal activation domain and the B domain that is required to interact with the SRF dimer.

**Figure 2 molecules-26-06125-f002:**
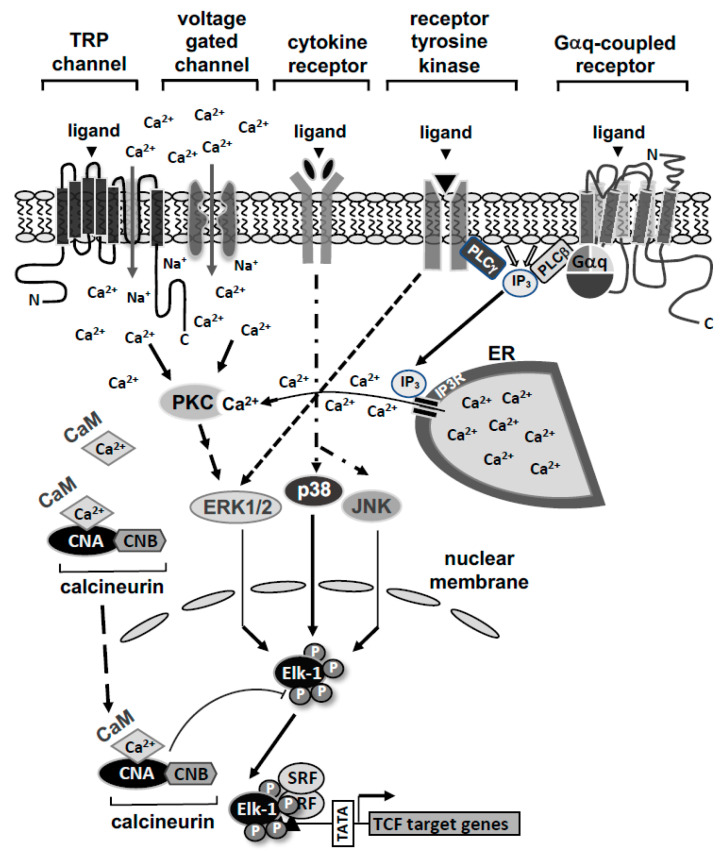
Stimulus-transcription coupling involving Elk-1. There are numerous receptors and ion channels expressed at the surface of mammalian cells that are necessary for the communication of the cells with their environment. Ligands of G protein-coupled receptors, receptor tyrosine kinases and cytokine receptors stimulate the cells as well as the activation of ligand- or voltage-gated ion channels (TRP channels, voltage-gated Ca^2+^ channels). Upon stimulation of the cells with extracellular signaling molecules, intracellular signaling cascades are initiated leading to the activation of the MAP kinases extracellular signal-regulated protein kinase (ERK1/2), c-Jun N-terminal protein kinase (JNK) and p38 protein kinase, which function as signal transducers. A rise in intracellular Ca^2+^ activates MAP kinases via stimulation of protein kinase C. MAP kinases translocate into the nucleus and phosphorylate Elk-1 and related TCF, and thus activate SRE-controlled gene transcription. Elk-1 is dephosphorylated by the Ca^2+^/calmodulin-dependent protein phosphatase calcineurin.

**Figure 3 molecules-26-06125-f003:**
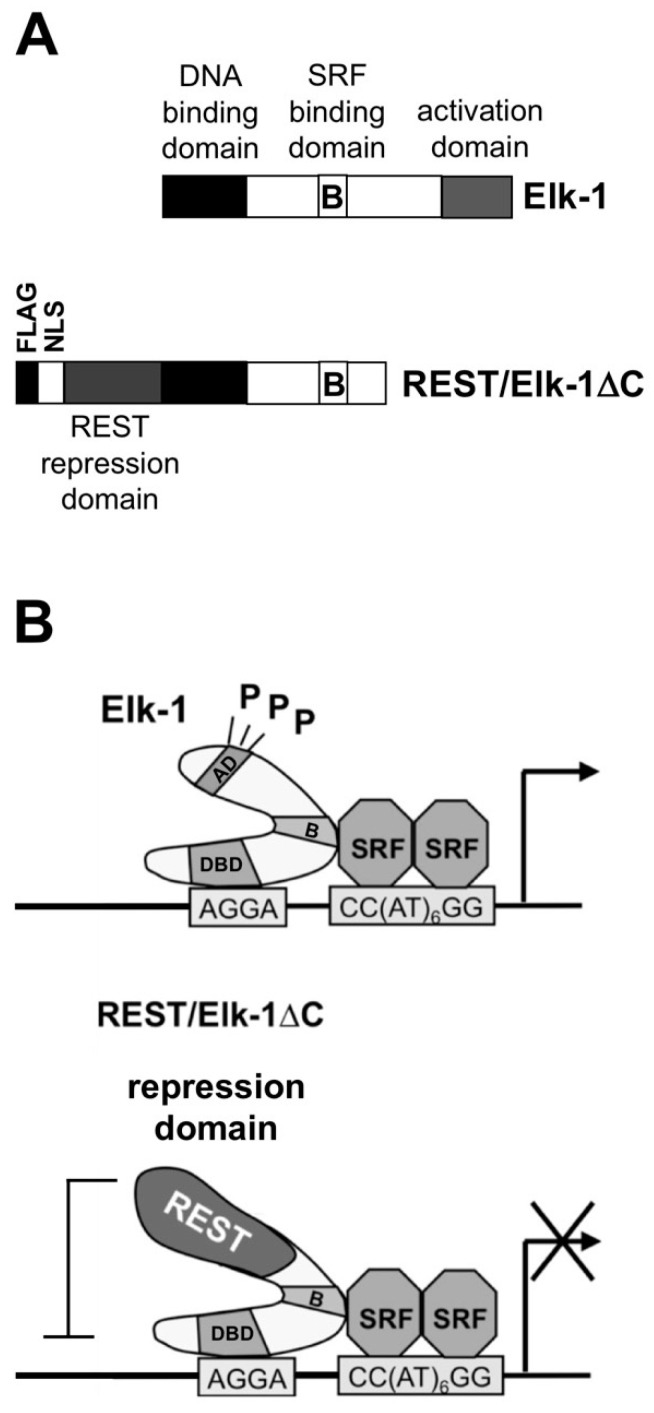
A dominant-negative mutant of Elk-1 allows the investigation of the biological function of Elk-1. (**A**) Modular structure of Elk-1 and REST/Elk-1ΔC, showing the N-terminal DNA binding domain, the SRF binding site (domain B) and the C-terminal transcriptional activation domain. The REST/Elk-1ΔC mutant lacks the activation domain, but contains instead the N-terminal transcriptional repressor domain from REST. (**B**) Mechanism of REST/Elk-1ΔC interference. The binding of TCF proteins to the SRE and to SRF is blocked. Additionally, histone deacetylases are recruited to the transcription units. P, Elk-1 phosphorylation site.

**Figure 4 molecules-26-06125-f004:**
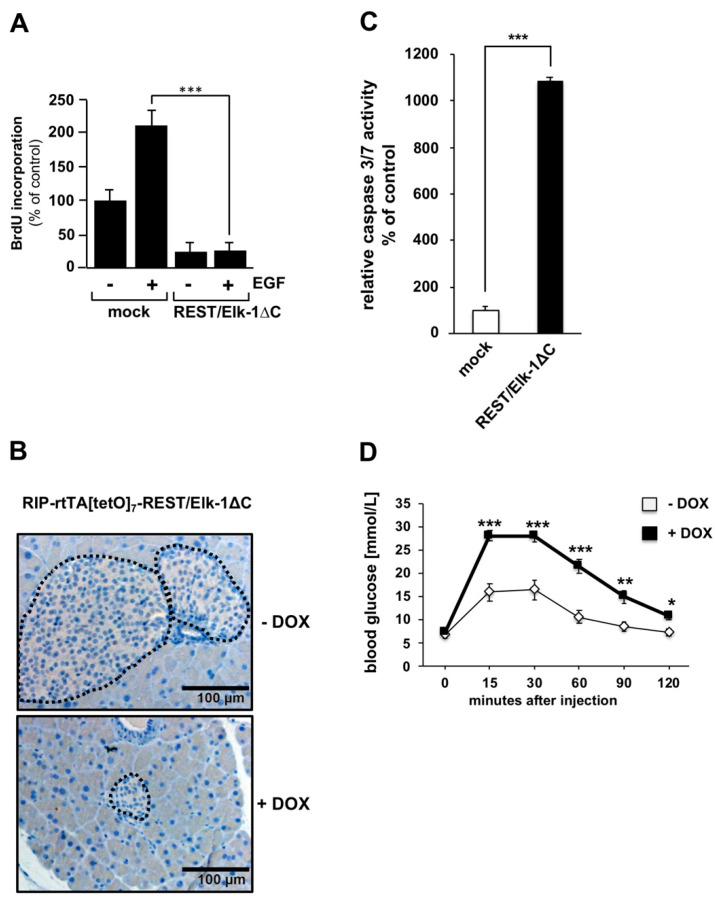
Biological functions of Elk-1. (**A**) Primary astrocytes were infected with a recombinant lentivirus encoding REST/Elk-1ΔC. As a control, mock-infected cells were analyzed. Cells were stimulated with EGF (10 ng mL^−1^) for 24 h. DNA synthesis was measured by the incorporation of BrdU into the DNA that was detected by an immunoassay (*** *p* < 0.001). (**B**) Attenuation of Elk-1 activity in pancreatic β-cells resulted in the generation of significantly smaller islets. Transgenic mice were analyzed that allowed the doxycycline (Dox)-dependent expression of the dominant-negative mutant of Elk-1, REST/Elk-1ΔC, specifically in pancreatic β-cells. H&E-stained sagittal sections of pancreata derived from these mice that had received doxycycline supplementation in the drinking water showed significantly smaller islets in comparison to control mice. (**C**) Inhibition of Elk-1 activated caspase-3/7. INS-1 832/13 insulinoma cells were infected with a recombinant lentivirus encoding REST/Elk-1ΔC as indicated. The cells were incubated in DME medium containing 0.5% serum and 2 mM glucose for 74 h. The caspase-3/7 activity was determined with the “CaspaseGlo-Substrate”. (**D**) Impaired glucose tolerance in transgenic mice expressing REST/Elk-1ΔC in pancreatic β-cells. A glucose tolerance test was performed with 10 week-old double-transgenic RIP-rtTA/[tetO]_7_REST/Elk-1ΔC mice that were maintained either in the presence or absence of doxycycline (DOX) in the drinking water. The animals were injected with glucose (2 g/kg body weight), and blood glucose levels were measured at different time points. Blood glucose concentrations were determined (data shown are mean +/− SEM or +/− SD; * *p* < 0.05; ** *p* < 0.01; *** *p* < 0.001; n = 3). Reproduced with modifications from Refs. [8,21], with permission from the Company of Biologists, John Wiley and Sons and Elsevier.

**Figure 5 molecules-26-06125-f005:**
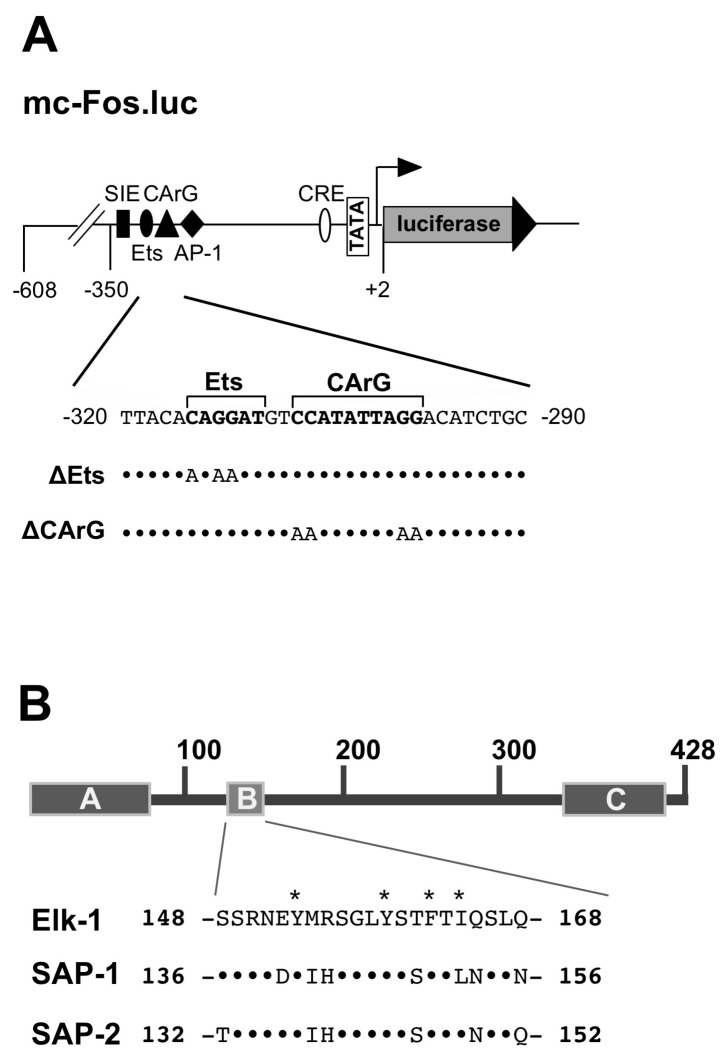
Role of DNA binding and protein–protein interaction to generate a functional ternary complex. (**A**) Sequence of the intact c-Fos promoter. The mutations to inactivate the DNA binding of either Elk-1 (ΔEts) or SRF (ΔCArG) are indicated. (**B**) Sequence alignment of the B-box of the TCF proteins Elk-1, SAP-1 and SAP-2. Alanine scanning identified the importance of the Elk-1 residues Y153, Y159, F162 and I164 (indicated with stars), while an X-ray analysis of a SAP-1-SRF complex showed that the SAP-1 residues Y141, L246, F150 and L155 contribute to the SRF binding interface. Reproduced with modifications from Refs. [25,28] with permission from John Wiley and Sons, and Elsevier. The sequence information is derived from Ref. [7].

**Figure 6 molecules-26-06125-f006:**
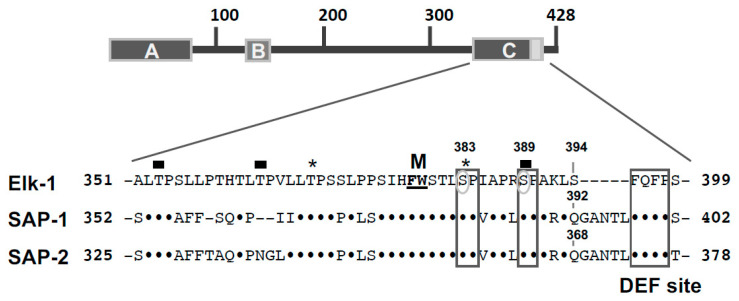
Phosphorylation sites within the activation domain of TCF proteins. Amino acid sequence of the transcriptional activation domains (C-box) of Elk-1, SAP-1 and SAP-2. The key phosphorylation sites S383 and S389 are highlighted. Rapid (stars) and intermediate phosphorylation sites (bars) are indicated. On the C-terminal end of the C-box is the second MAP kinase docking site (DEF site). In addition, the Med23 binding site (F378, W379) is shown (M). The sequence information is derived from Ref. [7].

**Figure 7 molecules-26-06125-f007:**
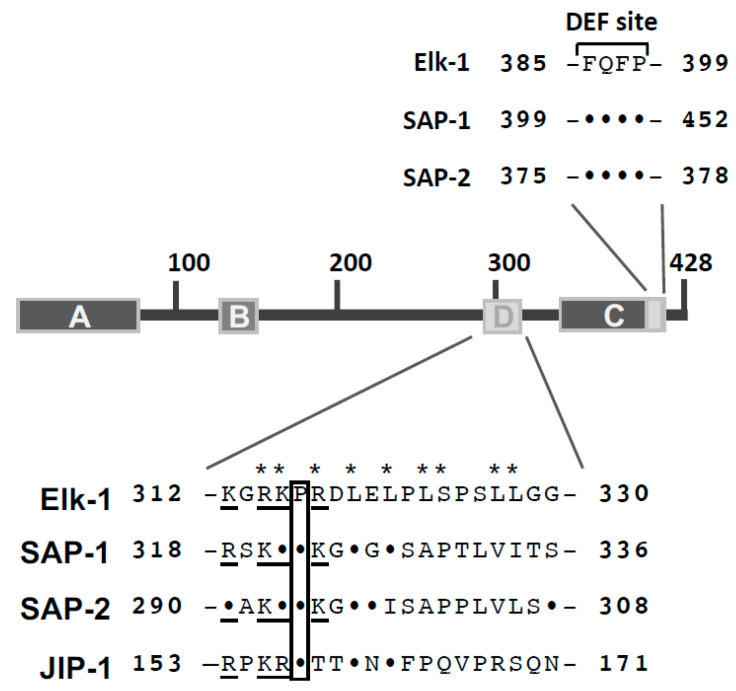
Protein kinase docking sites within the TCF proteins. Amino acid sequence of the D-box (D-domain, D-site) and the DEF site. The sequence information is derived from Ref. [7]. Critical amino acids of Elk-1 required for binding of MAP kinases are marked with stars. The sequence of the JNK docking site to the JNK-interacting protein-1 (JIP-1) is also depicted. The box shows a proline residue present in TCF proteins and JIP-1.

**Figure 8 molecules-26-06125-f008:**
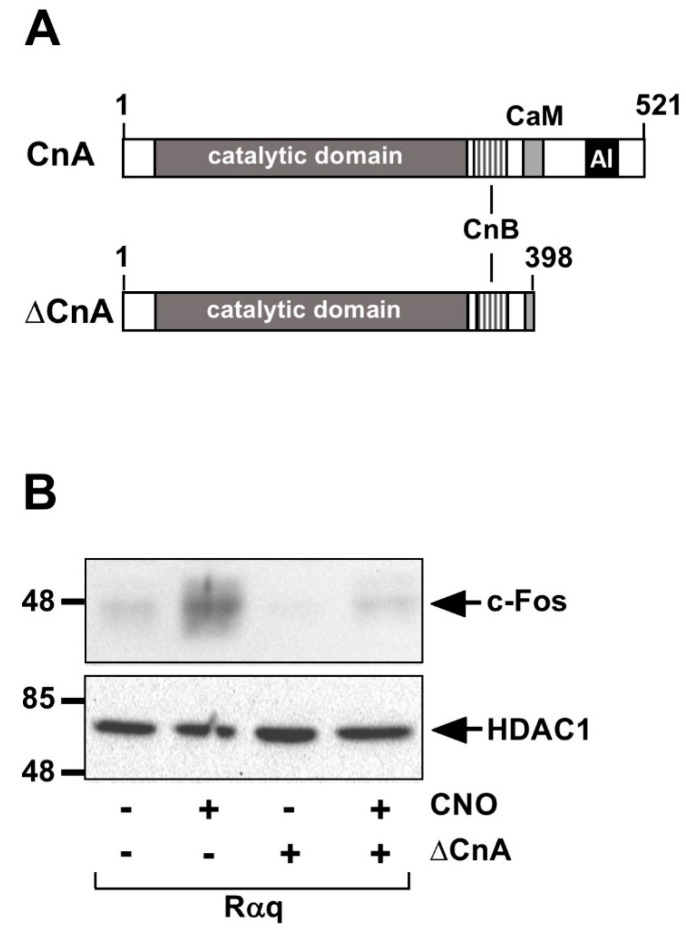
Calcineurin dephosphorylates and inactivates Elk-1. (**A**) Modular structure of calcineurin A and the truncated mutant ΔCnA, which is constitutively active because the autoinhibitory domain (AI) and a portion of the calmodulin binding site have been deleted. (**B**) Stimulation of the Gαq-coupled designer receptor (Rαq) with CNO activates c-Fos expression in HEK293 cells. Expression of ΔCnA attenuates receptor-induced c-Fos expression. HEK293 cells expressing Rαq were infected with a lentivirus encoding ΔCnA as indicated. Cells were cultured for 24 h in medium containing 0.05% serum and then stimulated with the Rαq ligand CNO (1 μM) for 3 h. The figure shows Western blot analysis of proteins of nuclear extracts using an antibody directed against c-Fos. The antibody directed against HDAC1 was used as a loading control. Reproduced with modifications from Ref. [57] with permission from Elsevier.

**Figure 9 molecules-26-06125-f009:**
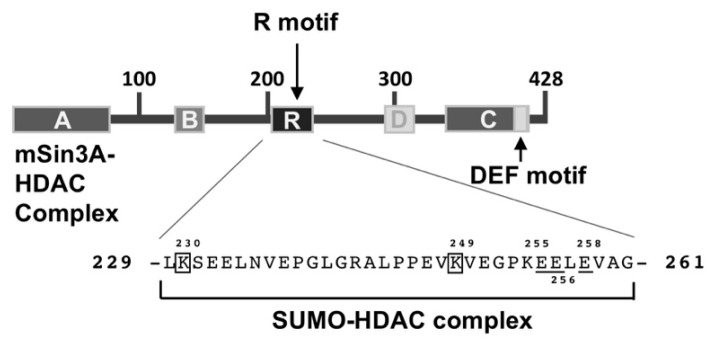
Elk-1 contains two distinct repressor domains. The N-terminal repressor domain recruits the corepressor protein mSin3A, complexed with histone deacetylases, to the N-terminus of Elk-1. The binding of both mSin3A and histone deacetylases is stimulated by the phosphorylation of Elk-1. The R motif represents a repression domain of Elk-1 that functions as a binding site for a SUMO–HDAC-2 complex. SUMO binding requires the lysine residues K230 and K249 (boxes) and a stretch of acidic residues (bars). The sequence information is derived from Ref. [7].

**Figure 10 molecules-26-06125-f010:**
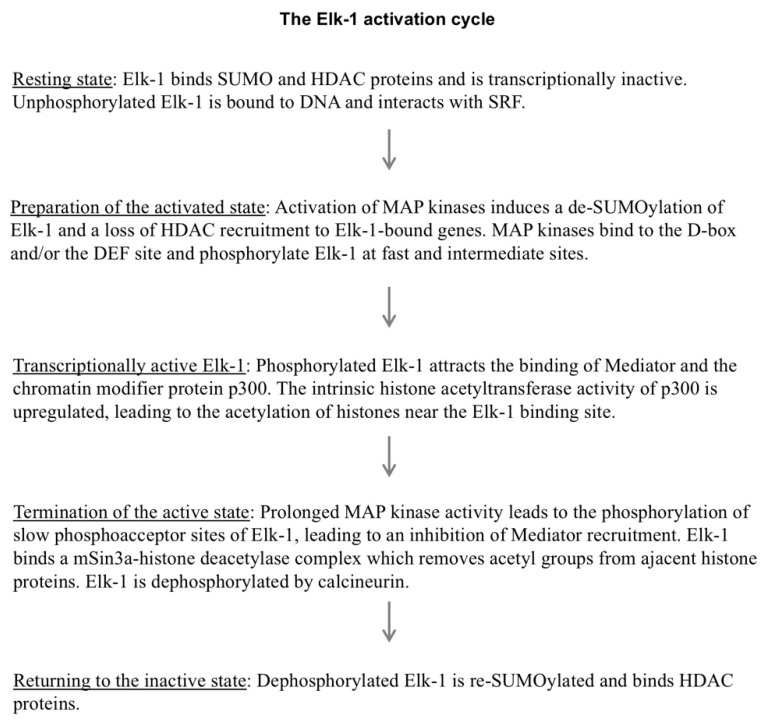
The Elk-1 activation cycle.

## Data Availability

Not applicable.

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
