# Peer review of "Critical Protein–Protein Interactions Determine the Biological Activity of Elk-1, a Master Regulator of Stimulus-Induced Gene Transcription"

_molecules, 2021, doi:10.3390/molecules26206125_

Round 1
Reviewer 1 Report
ELK-1 is a critical TF to modulate a number of physiological actions in human health and disease. This review summarized a comprehensive introduction of ELK-1 from the structure, function to the potential regulatory mechanisms. This will benefit further researches of ELK-1 in this field.
Comments:
1 ELK-1 belongs to Ets family; thus this family may be introduced briefly in the Background from start.
2 In the section of biological function of ELK-1, its functions in cancer research should be addressed.
3 A graphic scheme to conclude this protein-protein cross-talk may be added.
Author Response
September 6, 2021
Manuscript ID: molecules-1328919
I would like to thank the reviewer for the examination of our manuscript. We have dealt with the criticisms and comments and a new figure has been added. Please find below a point-by-point response to the points of criticism raised by the reviewer.
I hope that you find the revised version of the manuscript satisfactory and acceptable for publication in "Molecules“.
Sincerely yours,
Gerald Thiel, Ph.D.
___________________________________________________________________
ELK-1 is a critical TF to modulate a number of physiological actions in human health and disease. This review summarized a comprehensive introduction of ELK-1 from the structure, function to the potential regulatory mechanisms. This will benefit further researches of ELK-1 in this field.
Comments:
- ELK-1 belongs to Ets family; thus this family may be introduced briefly in the Background from start.
Response:
We have added the following sentences to Introduction:
Elk-1 is a member of the ETS family of transcription factors. The name ETS is derived from the avian erythroblastosis virus E26 which encodes the v-ets oncogene. Its cellular counterpart, Ets-1, is the founding protein of the ETS family of proteins. All family member bind to a similar DNA recognition site, encompassing the sequence GGAA/T, using their DNA binding site, the Ets domain.
- In the section of biological function of ELK-1, its functions in cancer research should be addressed.
Response:
We have added added the following sentences to the end of chapter 5:
Several proteins of the ETS family of transcription factors have been linked to tumor development and cancer [1]. The expression of v-ets, encoded by the E26 virus, results in the development of erythroid and myeloid leukemias. The ETS proteins PEA3 and ERG have been proposed to induce Ewing sarcomas and prostate cancer, respectively. Elk-1 has been proposed to play a role in different cancer types. However, the published studies do not allow for a general statement about a correlation between Elk-1 and tumor development.
- A graphic scheme to conclude this protein-protein cross-talk may be added.
Response:
We added a scheme as requested (Fig. 10).
Reviewer 2 Report
The review by Thiel and colleagues addresses the role of Elk1 protein-protein interactions in the modulation of its activity.
Overall, the manuscript is well written, but lacks relevant discussion on the data mentioned. The main purpose of writing a review is to create a readable synthesis of the best resources available in the literature for an important research question. However, the manuscript does not offer a focused discussion and often relevant literature citations are missing.
Moreover, the manuscript lacks a focused introduction, briefly placing the study in a broad context, and highlighting why the topic chosen is important.
The authors often show data they have already published, without discussing them critically. The overall take home message is not clear.
Author Response
September 6, 2021
Manuscript ID: molecules-1328919
I would like to thank the reviewer for the examination of our manuscript. We have dealt with the criticisms and comments. Please find below a point-by-point response to the points of criticism raised by the reviewer. I hope that you find the revised version of the manuscript satisfactory and acceptable for publication in "Molecules“.
Sincerely yours,
Gerald Thiel, Ph.D.
The review by Thiel and colleagues addresses the role of Elk1 protein-protein interactions in the modulation of its activity.
Overall, the manuscript is well written, but lacks relevant discussion on the data mentioned. The main purpose of writing a review is to create a readable synthesis of the best resources available in the literature for an important research question. However, the manuscript does not offer a focused discussion and often relevant literature citations are missing.
Response:
This manuscript focuses the sequential activation cycle of Elk-1 involving distinct protein-protein interactions that influences the basal transcriptional apparatus and the chromatin structure. Therefore, the manuscript fits very well into the special issue of “Molecules” entitled “Protein-Protein-Interactions 2021”.
There are several published review articles more broadly discussing ETS transcription factors (Sharrocks, 2001; Buchwalter et al., 2004; Hollenhorst et al., 2011), transcriptional regulation by MAP kinases (Yang et al., 2003; Yang et al., 2013), the serum response element (Treisman, 1992; Galbraith and Espinosa, 2011) and the role of Elk-1 in the brain (Besnard et al., 2011), but none of these reviews is focused on the essential role of protein-protein interactions within the Elk-1 activation cycle. Moreover, new data based on ChIP experiments are not addressed in these reviews.
Our article addresses controversial issues, e.g., the role of phosphorylation on Elk-1 DNA binding as well as the regulation of ternary complex formation. Moreover, we propose, based on articles by us and others, that protein-protein interaction between Elk-1 and SRF may be sufficient for generating a functional ternary complex that is bound only via the SRF to DNA. Additionally, the article puts a special emphasis on the involvement of chromatin modifiers within the Elk-1 activation cycle.
All relevant literature, including seminal articles from the Treisman, Sharrocks, Shaw, and Nordheim labs, is quoted in the article.
Moreover, the manuscript lacks a focused introduction, briefly placing the study in a broad context, and highlighting why the topic chosen is important.
Response:
The authors often show data they have already published, without discussing them critically. The overall take home message is not clear.
Response:
We have used data published by ourselves and others to illustrate the biological function and interactions of Elk-1. The previously published work showed the importance of Elk-1 as a major regulator of cellular proliferation and cell death. The focus of this article is not the biological function of Elk-1 but rather the sequential activation cycle of Elk-1 which involves numerous protein-protein interactions. We have included a statement regarding the focus of the article in the introduction. As requested by the other reviewer we have added a schematic for the protein-protein cross-talk. We are hopeful this clarifies our overall message.
Round 2
Reviewer 1 Report
This revised version is accepted by this reviewer.
Author Response
The reviewer did not request further modifications of the manuscript.
Reviewer 2 Report
Major criticisms raised in the first round of revision have not been adequately addressed.
Author Response
I would like to thank the reviewer for the examination of our manuscript. We have dealt with the criticisms and comments. We addressed all points raised by reviewer # 2 in our revised manuscript. Please find below a point-by-point response to the points of criticism raised by the reviewer
- Reviewer # 2 stated that the article “lacks relevant discussion on the data mentioned.” We disagreed with this statement and pointed out in the rebuttal letter that important questions concerning the molecular biology of Elk-1 are discussed in the article. This includes the importance of Elk-1 DNA binding as well as the importance of Elk-1 phosphorylation for DNA binding and ternary complex formation. In particular, we discussed that protein-protein interaction between Elk-1 and SRF may be sufficient for generating a functional ternary complex that is bound only via the SRF to DNA, indicating that Elk-1 DNA binding is of lesser importance. As part of our manuscript, new research data are included that were not available for the authors of previous review articles about Elk-1. We included new results based on chromatin immunoprecipitation experiments in the description of protein-protein interactions of Elk-1. A special emphasis was put on the involvement of chromatin modifiers within the Elk-1 activation cycle.
- Reviewer # 2 stated that “the manuscript does not offer a focused discussion”. The intention of the review article, as outlined in the “Introduction” section, was the description of the sequential activation cycle of Elk-1 involving distinct protein-protein interactions that influence the basal transcriptional apparatus and the chromatin structure. The article clearly states the importance for protein-protein interactions for the biological activity of Elk-1 and the stimulus-induced activation cycle of Elk-1. Previous published review articles about Elk-1 discussed ETS transcription factors (Sharrocks, 2001; Buchwalter et al., 2004; Hollenhorst et al., 2011), the transcriptional regulation by MAP kinases (Yang et al., 2003; Yang et al., 2013), the serum response element (Treisman, 1992; Galbraith and Espinosa, 2011), or the role of Elk-1 in the brain (Besnard et al., 2011), but none of these review articles focused on the essential role of protein-protein interactions within the Elk-1 activation cycle. Our review article is focused on specific protein-protein interactions in the activation cycle of Elk-1. It is distinct from previous review articles about Elk-1 and would fit very well into the special issue of “Molecules” entitled “Protein-Protein-Interactions 2021”.
- Reviewer # 2 stated that “often relevant literature citations are missing.” To our knowledge the review provides a fair and unbiased coverage of the relevant literature. All relevant literature, including seminal articles from the Treisman, Sharrocks, Shaw, and Nordheim labs, is quoted in the article. The reviewer did not specify which literature citations they feel are missing in the review article. We would have been happy to include these additional articles if they indeed fit into the text.
- Reviewer # 2 stated that the “authors often show data they have already published, without discussing them critically. The overall take home message is not clear.” The data shown in this review article illustrate the biological functions and interactions of Elk-1 which is necessary for the readers that are not working in this field to understand the importance of Elk-1 as a major regulator of cellular proliferation and cell death. The focus of this article is not the biological function of Elk-1 but rather the sequential activation cycle of Elk-1 which involves numerous proteins-protein interactions. We provided this context in the “Introduction” section.